# BCG Vaccination-Associated Lower HbA1c and Increased CD25 Expression on CD8^+^ T Cells in Patients with Type 1 Diabetes in Ghana

**DOI:** 10.3390/vaccines12050452

**Published:** 2024-04-24

**Authors:** Wilfred Aniagyei, Sumaya Mohayideen, Osei Sarfo-Kantanka, Sarah Bittner, Monika M. Vivekanandan, Joseph F. Arthur, Agnes O. Boateng, Augustine Yeboah, Hubert S. Ahor, Shadrack O. Asibey, Elizabeth Owusu, Diran Herebian, Maximilian Huttasch, Volker Burkart, Robert Wagner, Michael Roden, Ernest Adankwah, Dorcas O. Owusu, Ertan Mayatepek, Marc Jacobsen, Richard O. Phillips, Julia Seyfarth

**Affiliations:** 1Department of General Pediatrics, Neonatology and Pediatric Cardiology, Medical Faculty and University Hospital Düsseldorf, Heinrich Heine University Düsseldorf, 40225 Düsseldorf, Germany; 2Kumasi Centre for Collaborative Research in Tropical Medicine (KCCR), Kumasi 00233, Ghanajrkwaku2016@gmail.com (A.Y.); dorcaso.owusu@gmail.com (D.O.O.);; 3Komfo Anokye Teaching Hospital, Kumasi 00233, Ghana; 4School of Medicine and Dentistry, College of Health Sciences, Kwame Nkrumah University of Science and Technology (KNUST), Kumasi 00233, Ghana; 5Institute for Clinical Diabetology, German Diabetes Center, Leibniz Center for Diabetes Research at Heinrich Heine University Düsseldorf, 40225 Düsseldorf, Germany; 6German Center for Diabetes Research, Partner Düsseldorf, 85764 Neuherberg, Germany; 7Department of Endocrinology and Diabetology, Medical Faculty and University Hospital Düsseldorf, Heinrich Heine University Düsseldorf, 40225 Düsseldorf, Germany

**Keywords:** type 1 diabetes, BCG vaccination, immunomodulation, T cells, glycolysis

## Abstract

BCG vaccination affects other diseases beyond tuberculosis by unknown—potentially immunomodulatory—mechanisms. Recent studies have shown that BCG vaccination administered during overt type 1 diabetes (T1D) improved glycemic control and affected immune and metabolic parameters. Here, we comprehensively characterized Ghanaian T1D patients with or without routine neonatal BCG vaccination to identify vaccine-associated alterations. Ghanaian long-term T1D patients (*n* = 108) and matched healthy controls (*n* = 214) were evaluated for disease-related clinical, metabolic, and immunophenotypic parameters and compared based on their neonatal BCG vaccination status. The majority of study participants were BCG-vaccinated at birth and no differences in vaccination rates were detected between the study groups. Notably, glycemic control metrics, i.e., HbA1c and IDAA1c, showed significantly lower levels in BCG-vaccinated as compared to unvaccinated patients. Immunophenotype comparisons identified higher expression of the T cell activation marker CD25 on CD8^+^ T cells from BCG-vaccinated T1D patients. Correlation analysis identified a negative correlation between HbA1c levels and CD25 expression on CD8^+^ T cells. In addition, we observed fractional increases in glycolysis metabolites (phosphoenolpyruvate and 2/3-phosphoglycerate) in BCG-vaccinated T1D patients. These results suggest that neonatal BCG vaccination is associated with better glycemic control and increased activation of CD8^+^ T cells in T1D patients.

## 1. Introduction

The Bacillus Calmette–Guérin (BCG) vaccine is one of the most common global vaccines. It is utilized in several countries worldwide for the prevention of severe tuberculosis manifestations, especially in regions with a high prevalence of tuberculosis. Recent studies have indicated potential off-target benefits of BCG, including reduced mortality among vaccinated individuals [1,2]. Specifically, BCG vaccination may offer general protection from respiratory tract infections and other viral and bacterial diseases [3,4]. Most recently, an effect of the BCG vaccination on COVID-19 diseases was discussed [5,6,7]. The mechanisms of these off-target effects of BCG vaccination are still under investigation.

Recent research has increasingly focused on the potential impact of BCG vaccination on autoimmune diseases, including type 1 diabetes (T1D). The reason for this emanated from studies conducted in animal models which indicated that BCG vaccination prevents the development of T1D [8]. In humans, however, BCG vaccination was not associated with differences in the incidence of T1D [9,10]. Nevertheless, recent studies suggested that the administration of BCG vaccines in long-term T1D patients influenced the disease course of T1D. Vaccinated patients showed a long-term reduction in hyperglycemia with near-normal levels of HbA1c after three years [11]. These studies have suggested possible explanations, such as increased Treg signatures and enhanced aerobic glycolysis, that may contribute to the protective effects of BCG against T1D [12,13]. However, other studies showed conflicting results, indicating marginal or no beneficial effect of BCG vaccination on T1D [14,15]. It is also not known whether neonatal BCG vaccination, in contrast to BCG vaccination in overt diabetes as previously studied [11,14,15], can influence the clinical course of T1D disease.

In Ghana, like in most Sub-Saharan African countries, neonatal BCG vaccination is recommended and administered to healthy neonates, although a subgroup remains unvaccinated [16]. T1D is one of the most frequent non-communicable chronic diseases in Ghanaian children, and the number of adolescents diagnosed with T1D has increased over the last decades [17]. This provides a unique opportunity to study the long-term effects of BCG vaccination in newborns on the later-life disease course of T1D. For this purpose, we compared T1D-relevant clinical, immune, and metabolic parameters between matched BCG-vaccinated and unvaccinated T1D patients. Healthy controls were included to compare vaccination rates between patients and controls and to assess vaccine-associated but non-T1D-specific alterations. Correlations between immune and clinical parameters were performed to elucidate the underlying mechanisms.

## 2. Materials and Methods

### 2.1. Study Cohort Characterization

This cross-sectional cohort study comprised structured interviews, non-fasting whole blood and serum sampling, and laboratory testing. T1D patients (*n* = 108) and healthy controls (*n* = 214) were recruited between August 2021 and January 2023 at the Diabetes Clinic, Komfo Anokye Teaching Hospital, the main referral clinic for diabetes in the middle belt of Ghana. Diagnosis of T1D followed the American Diabetes Association criteria [18]. In brief, laboratory tests determined either a fasting blood glucose concentration of ≥7.0 mmol/L or an increased blood glucose concentration during an oral glucose tolerance test (OGTT). Confirmatory testing at onset of disease using T1D autoantibody screening and C-peptide measurement was performed in ambiguous cases. Patients with other forms of diabetes, auto-immune diseases other than T1D, and active tuberculosis were excluded from this study. Only T1D patients with persistent insulin dependency were included. The control group comprised individuals with a negative history of autoimmune or systemic inflammatory diseases or diabetes. No pregnant or breastfeeding women or participants with active COVID-19 symptoms or positive COVID-19 tests were included in this study. Study group characteristics are given in Appendix A. This study was approved by the Ethics Committee Board of the Komfo Anokye Teaching Hospital in Kumasi, Ghana. All participants and their legal guardians provided written informed consent.

### 2.2. BCG Vaccination

As recommended by the WHO for countries with a high incidence of TB, the BCG vaccination policy in Ghana is a one-time administration as soon as possible after birth [19]. Booster vaccination is not recommended [19]. Precise information on the BCG strain used for each patient is not available. General information on the BCG strains used in the routine vaccination program was summarized in a previous observational study [20]. All study participants were screened for the presence of a BCG vaccination scar which is a key feature after vaccination [21]. BCG scar-positive patients were classified as BCG-vaccinated.

### 2.3. Serum Measurements

The serum samples were thawed simultaneously from −80 °C and used for the analysis of serum parameters.

#### 2.3.1. Random C-Peptide Measurement

C-peptide determinations were performed using the e801 module of a Cobas 8000 system (Roche Diagnostics Deutschland GmbH, Mannheim, Germany). Serum samples of T1D patients were assayed for C-peptide using Electrochemiluminescence Immunoassay (ECLIA).

#### 2.3.2. T1D Autoantibodies

Glutamic acid decarboxylase (GAD) antibodies were measured by a radioligand assay [22] (cutoff 2 U/mL), and the islet cell autoantibodies IA-2 and ZnT8 were measured by a radioimmunoassay (cutoff 2 U/mL) and an enzyme-linked immunosorbent assay (cutoff ≥ 15 U/mL) [23], respectively (Medipan GmbH, Dahlewitz, Germany).

#### 2.3.3. Analysis of Targeted Metabolites by Liquid Chromatography–Tandem Mass Spectrometry (LC-MS/MS)

Serum sample preparation was carried out by protein precipitation with methanol-acetonitrile (1:1, *v*/*v*) containing isotopically labeled internal standards. The targeted compounds were analyzed by ultra-high-performance liquid chromatography-MS/MS (UPLC-MS/MS). The system consists of a UPLC I-Class (Waters, Milford, MA, USA) coupled to a tandem mass spectrometer Xevo-TQ-XS (Waters, Wilmslow, UK). Electrospray ionization was performed in the negative ionization mode for the compounds fructose-1,6-bisphosphate, phosphoenolpyruvate (PEP), 2/3-phosphoglycerate, lactate, and pyruvate as described in the literature [24]. The analytes N2, N2-dimethylguanosine, and N6-carbamoylthreonyladenosine were measured in the positive ionization mode as previously described [25]. Mass spectrometric quantitation of the compounds was carried out in the multiple reaction monitoring (MRM) mode. MassLynx software (v4.2; Waters, Milford, MA, USA) was used for instrument control and data acquisition. Quantitation analysis was performed by TagetLynx XS software (Waters, Milford, MA, USA).

### 2.4. T Cell Phenotyping by Flow Cytometry

Heparinized whole blood (10 mL) was collected from all study participants. PBMCs were enriched from blood samples by density gradient centrifugation (Histopaque-1077; Sigma-Aldrich, St. Louis, MO, USA) following the manufacturer’s guidelines. Without cryopreservation or batching, isolated PBMCs (2 × 10^5^) from all study subjects were stained for viability (1:100; viability dye-e780, eBioscience/Thermo, San Diego, CA, USA) and the following fluorescence-labeled anti-human antibodies: CD8 (FITC, HIT8a; BioLegend, San Diego, CA, USA), CD3 (AF700, SK7; BioLegend), CD127 (PE-Cy7, AO19D5; BioLegend), CD25 (BV421, BC96; BioLegend), and CD4 (BV510, RPA-T4; BioLegend). Cells were incubated for 30 min on ice in the dark, washed twice (100 µL D-PBS Sterile, Gibco, Gaithersburg, MD, USA), resuspended in 100 µL D-PBS Sterile (Gibco), and acquired with a 13-color Beckman Coulter CytoFLEX S flow cytometer, equipped with 4 lasers. Fluorescence minus one (FMO) control for anti-human CD25 (BV421) and anti-human CD127 (PE-Cy7) was included to identify the respective background. A minimum of 1 × 10^5^ viable cells was recorded for each sample. FACS data were exported and analyzed with the FlowJo software (Version 10, FlowJo LLC, Ashland, OR, USA).

### 2.5. Statistical Analysis

Statistical analyses were conducted using GraphPad Prism v9 software (GraphPad Software, La Jolla, CA, USA) and IBM SPSS Statistics (Version 29). All figures were prepared using GraphPad Prism. Non-parametric tests were employed as the data did not follow a normal distribution (confirmed by Shapiro–Wilk and Kolmogorov–Smirnov tests). The Mann–Whitney U test was utilized for study group comparisons. Spearman rank correlation coefficient was utilized to evaluate the relationship between HbA1c and T cell phenotype marker expression. Categorical variables were compared between the study groups using Fisher’s exact test. Statistical significance was defined as a *p*-value below 0.05.

## 3. Results

### 3.1. Study Characteristics and BCG Vaccination Rates

We recruited T1D patients (*n* = 108) and age- and sex-matched healthy controls (*n* = 214) (for demographic characteristics, see Appendix A). All participants were investigated for indications of neonatal BCG vaccination via the presence of a BCG vaccination scar which is a key feature after vaccination [21]. The majority of both T1D patients and controls were classified as BCG-vaccinated (T1D: *n* = 90, 83.3%; healthy controls: *n* = 165, 77.1%). There was no significant difference in the BCG vaccination rate between T1D patients and controls (*p* = 0.1935; Figure 1A). We also compared the BCG-vaccinated subgroups of T1D patients and controls for age, sex, and BMI distributions. These comparisons did not reveal significant differences between vaccinated and unvaccinated patients or controls (Appendix A).

### 3.2. BCG-Vaccinated T1D Patients Had Lower HbA1c Compared to Unvaccinated Patients

To determine the potential effect of BCG vaccination on T1D disease, we compared clinical parameters between BCG-vaccinated (*n* = 90) and unvaccinated T1D patients (*n* = 18). There were no differences in terms of onset age (*p* = 0.0953) and duration of disease (*p* = 0.7408; Figure 1B). We also measured the T1D-associated autoantibodies against GAD65, IA-2, and ZnT8 for all T1D patients. There were no differences between the BCG-vaccinated and unvaccinated groups in any of the T1D-associated autoantibodies (Figure 1C).

Next, we compared diabetes-relevant metrics (C-peptide and HbA1c as surrogate markers of ß-cell function and glycemic control, respectively) between BCG-vaccinated and unvaccinated patients. While the two groups did not differ in C-peptide concentrations (*p* = 0.3647, Figure 1D), the HbA1c levels were lower in the BCG-vaccinated group (*p* = 0.0057, Figure 1D). To investigate whether a shorter or longer vaccination history is associated with differences in HbA1c, we determined the correlation between the age of the vaccinated patients and their HbA1c. No correlation was observed (Appendix A). Differential HbA1c values may be influenced by the individual insulin intake of T1D patients, but neither the daily insulin dosages nor daily insulin dosages per weight differed between the BCG-vaccinated and unvaccinated groups (Figure 1E). In addition, we calculated the Insulin-Dose Adjusted A1c (IDAA1c) using HbA1c (%) + [4 × insulin dose (IU/kg per day)] to take into consideration both insulin requirements and HbA1c levels in a single value. We found that IDAA1c values were significantly lower in BCG-vaccinated T1D patients (*p* = 0.0136, Figure 1F).

These results indicated better glycemic control in T1D patients who were BCG vaccinated as newborns while there was no evidence for effects of neonatal BCG vaccination on T1D susceptibility.

### 3.3. CD25 Expression on CD8^+^ T Cells Is Higher in BCG-Vaccinated T1D Patients

Previous studies attributed the influence of BCG vaccination to immunological as well as metabolic effects [11]. The immune effects include a systemic impact on regulatory T cells, suggesting a role in restoring immune balance [13,26,27]. Hence, we next characterized the regulatory T cell phenotype in age- and sex-matched sub-cohorts of BCG-vaccinated (*n* = 17) and unvaccinated T1D patients (*n* = 17) (for demographic characteristics of the sub-cohorts, see Appendix A). Figure 2A depicts the full gating strategy and an example of synchronized Treg gating based on CD25 (Interleukin 2 receptor alpha) and CD127 (Interleukin 7 receptor alpha) expression as commonly described [28]. We compared BCG-vaccinated and unvaccinated T1D patients for the proportions of Treg in CD4^+^ and CD8^+^ T cells. There were no differences in the frequencies of CD4^+^ and CD8^+^ Tregs between the two groups. Also, we measured the protein expression level of CD25 on CD4^+^ and CD8^+^ T cells and compared the two groups. While CD25 expression within the CD4^+^ T cells did not differ, higher expression of CD25 was detected within CD8^+^ T cells of BCG-vaccinated T1D patients (*p* = 0.0187; Figure 2B). The same measurements of CD25 marker expression on T cells were performed for an age- and sex-matched vaccinated and unvaccinated healthy control cohort (Appendix A). No differences were detected between BCG-vaccinated (*n* = 17) and unvaccinated controls (*n* = 17) (Appendix A).

**Figure 2 vaccines-12-00452-f002:**
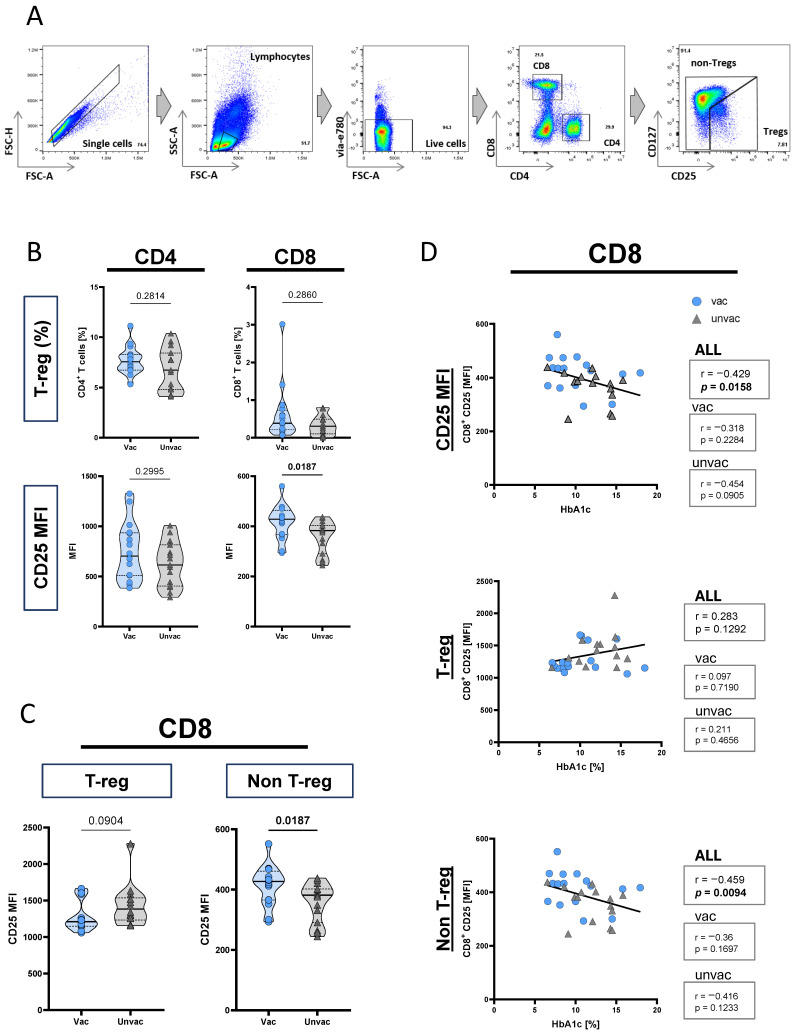
Comparison of T cell subsets and CD25 expression between BCG-vaccinated and unvaccinated T1D patients. Ex vivo phenotyping of peripheral blood lymphocytes from age- and sex-matched BCG-vaccinated T1D patients (circles, *n* = 17) and unvaccinated T1D patients (triangles, *n* = 17) is shown. Each symbol represents the mean of duplicates from an individual donor. (**A**) A representative depiction of the gating procedure for T cell subsets (i.e., CD4^+^ and CD8^+^) is provided. (**B**) Proportions of regulatory T cells and expression of CD25 MFI from CD4^+^ and CD8^+^ T cells are shown as violin plots including 25, 50, and 75 percentiles (as dotted or straight lines). (**C**) CD25 expression is shown as violin plots for non-regulatory and regulatory subsets of CD8^+^ T cells. Study group comparisons were performed, and *p*-values were calculated using the two-tailed Mann–Whitney U test. (**D**) Correlation plots for HbA1c and CD25 MFI of CD8^+^ T cells are shown. The Spearman rank test was applied to determine correlations for all patients and vaccination groups separately. A trend line was fitted by linear regression analysis. Correlation coefficients (r) and nominal *p*-values (*p*) are given. Age- and sex-matched BCG-vaccinated T1D patients (circles, *n* = 16) and unvaccinated T1D patients (triangles, *n* = 15) are shown. MFI, mean fluorescence intensity.

Since CD25 is both a marker of regulatory T cells and also of activated effector T cells [29], we next measured the protein expression level of CD25 on the Treg and non-Treg arms of CD8^+^ T cells. Within CD8^+^ Tregs, CD25 expression did not differ between vaccinated and unvaccinated patients (*p* = 0.09). In contrast, CD8^+^ non-Tregs from BCG-vaccinated T1D patients expressed significantly higher levels of CD25 (*p* = 0.0187; Figure 2C). There were no differences in CD25 expression on either Treg or non-Treg arms of CD8^+^ T cells between BCG-vaccinated and unvaccinated controls (Appendix A). Taken together, these results indicated that BCG-vaccinated T1D patients were characterized by CD25 high-expressing CD8^+^ T cell proportions, and this CD25 expression was enriched exclusively on the non-Treg subset of CD8^+^ T cells.

### 3.4. CD25 Expression on CD8^+^ T Cells Correlates with HbA1c in T1D Patients

To further characterize the effect of BCG vaccination on both HbA1c and CD8^+^ CD25 expression in T1D patients, we examined the possible connection between these two parameters. To this end, we correlated HbA1c levels with CD25 expression on T cells and subpopulations of Tregs and non-Tregs. Notably, an inverse correlation between HbA1c and CD25 protein expression of CD8^+^ T cells was detected (*p* = 0.0158, r = −0.429; Figure 2D). Next, we investigated the origin of the negative correlation between HbA1c and CD8^+^ CD25 expression. We split the CD25 expression on CD8^+^ T cells into Treg and non-Treg subsets and found that only CD25 expression on CD8^+^ non-Treg correlated negatively and significantly with HbA1c levels (*p* = 0.0094, r = −0.459; Figure 2D). On the contrary, CD25 expression on CD8^+^ Treg showed no correlation with HbA1c (*p* = 0.1292, r = 0.283; Figure 2D). To further investigate a general association between CD25 expression and glycemic control, we also correlated CD25 expression on CD4^+^ T cells with HbA1c in patients. Indeed, a significant negative association was observed between HbA1c and CD25 expression on CD4^+^ T cells (*p* = 0.0006, r = −0.579; Appendix A). Interestingly, this negative correlation was seen in the Treg (*p* = 0.0069, r = −0.475) and non-Treg (*p* = 0.0009, r = −0.566) subsets of CD4^+^ T cells, confirming the negative association between CD25 expression and glycemic control. Put together, these results indicate that the increased CD25 expression of non-Tregs in vaccinated patients is associated with better glycemic control in T1D patients.

### 3.5. Glucose Metabolites Differentiate BCG-Vaccinated Patients from Unvaccinated Patients

Another hypothesis for improved glycemic control in BCG-vaccinated long-term T1D suggests a systemic switch in glucose metabolism from oxidative phosphorylation to aerobic glycolysis [11,12]. In particular, Kühtreiber et al. found an increase in intermediates of de novo purine synthesis and glycolysis in T1D patients who had received BCG vaccination after disease onset [11]. Since changes in the phenotype and activation of T cells, as we have found, also frequently go hand in hand with altered glucose utilization [30,31], we investigated serum levels of metabolites and intermediates of glucose-processing pathways as well as intermediates of purine synthesis and compared T1D patients with and without (*n* = 17) neonatal BCG vaccination.

As intermediates of de novo purine synthesis, N6-carbamyolthreonyladenosine and N2, N2-dimethylguanosine were measured. No significant differences were found between vaccinated and unvaccinated patients (Figure 3A). To determine whether glucose metabolism parameters are altered in vaccinated T1D patients, we measured the intermediates of glycolysis fructose-1,6-bisphosphonate (F-1,6-BP), phosphoenolpyruvate (PEP), and 2/3-phosphoglycerate (2/3-PG) [32], as well as lactate and pyruvate, the end products of glycolysis under anaerobic and aerobic conditions, respectively [33]. While lactate and pyruvate levels were similar in both vaccinated and unvaccinated patients (Figure 3B), the glycolytic intermediates PEP (*p* = 0.0274) and 2/3-PG (*p* = 0.0180) were increased in vaccinated T1D patients (Figure 3C). These results indicated a partial increase in glycolytic intermediate metabolites, which may reflect changes in glucose utilization in vaccinated T1D patients.

## 4. Discussion

This study provided evidence that neonatal BCG vaccination may influence the clinical course of T1D. Lower HbA1c and IDAA1c levels in vaccinated T1D patients were accompanied by an increased CD25 expression on CD8^+^ T cells and a partial increase in glycolysis intermediates, suggesting an underlying role of T cell phenotypes and glucose utilization.

This is, to the best of our knowledge, the first study describing the impact of neonatal BCG vaccination on the clinical course of T1D developing later in life. Besides a general negative association between the occurrence of T1D and tuberculosis incidence at population level [34], several previous studies have shown that neonatal BCG vaccination has no effect on the development of T1D [9,35,36]. This was confirmed by our data, as vaccination rates did not differ between T1D patients and healthy controls. Since the BCG vaccination is administered only as a single vaccination in Ghana, we were unable to investigate the influence of the number of vaccinations on T1D, as has been done in other studies [37]. BCG vaccination in our cohort did not affect age at disease onset, in contrast to what was observed by others [36,38]. However, we observed an association of neonatal BCG vaccination with improved glycemic control after the development of T1D with significantly lower HbA1c and IDAA1c levels in vaccinated T1D patients. An influence of BCG vaccination on glycemic control has so far been shown for vaccinations applied to long-term T1D patients and the clinical effects could only be observed after several years [11]. For vaccination of new-onset T1D patients, such an effect was not observed in previous studies, although it should be noted that these patients were only followed up for a maximum of 2 years [15,39]. Our cohort of long-term T1D patients had a median disease duration of 6 years, and the time between BCG vaccination after birth and enrollment in our study averaged more than 20 years, suggesting that any effect mediated by BCG vaccination is very durable. In line with this, there was also no correlation between the age of the patients (corresponding to the period after vaccination) and HbA1c, which does not indicate a waning effect of the vaccination over time. In previous studies, lower HbA1c levels were associated not only with BCG vaccination but also with BCG treatment for bladder cancer in elderly patients [40]. This effect was observed specifically in T1D and not in type 2 diabetes patients and also lasted for many years [40].

The underlying immune or metabolic mechanisms that led to better glycemic control in T1D patients who received the vaccine after the onset of the disease are not fully understood. Kühtreiber et al. suggest a shift of immune balance towards regulatory T cells (Treg) thus affecting the disease pathomechanism [11]. In their studies, epigenetic induction of Treg signature genes resulted in increased Treg gene expression and Treg populations in BCG-vaccinated individuals [13]. These findings suggest that BCG can potentially restore or promote Treg activity, which is crucial for immune balance and may have clinical applications in combating T1D. In the present study, we did not observe differences in Treg proportions between vaccinated and unvaccinated patients. Accordingly, we and others did not find differential C-peptide levels as surrogate markers of insulin secretion [11,41]. This suggests that while BCG-related immune modulation can have a long-term impact on blood glucose control, it may not directly influence the destruction of ß-cells.

Although we saw similar proportions of Tregs, we found increased CD25 expression on CD8^+^ T cells from vaccinated T1D patients. CD25 plays a dual role as a regulatory T cell marker as well as an activation T cell marker. While being highly expressed on Tregs, CD25 expression on T cells, in general, indicates immune activation and has been associated with autoimmune diseases [28,29,42]. Following this, we found that increased CD25 expression was exclusive to non-Tregs and not in the Tregs subset of CD8^+^ T cells. This argues in favor of CD25 as an activation rather than regulatory marker in this context. Increased immune cell activation of both CD4^+^ and CD8^+^ T cells after BCG vaccination has been reported by other studies [43]. Both surface marker expression and cytokine production of CD4^+^ and CD8^+^ T cells were shown to be affected [44,45]. This heterologous lymphocyte response is thought to be crucial for the enhanced immune response to unrelated infections induced by BCG vaccination [46]. Our findings indicate that increased expression of activation marker CD25 on CD8^+^ T cells may be a long-term effect of vaccine-associated immune reprogramming in T1D patients. CD25 expression is essential for the optimal expansion of CD8^+^ T cells, and its upregulation is influenced by CD4^+^ T cells [47]. CD25-expressing non-regulatory CD8^+^ T cells have been identified as potent memory cells of great diversity, indicating their significance in immune responsiveness [48].

Interestingly, we found a direct negative correlation between CD25 expression and HbA1c suggesting that the activation status is directly linked to improved glycemic control in vaccinated T1D patients. Our findings are supported by other studies that show a similar relationship between glucose utility and immune activation and function [30,31,49]. In our study, the correlation of CD25 and HbA1c was found not only for both vaccinated and unvaccinated patients’ CD8^+^ T cells but also for CD4^+^ T cells, indicating a general and not-vaccine-specific association. Taken together, these data suggest that an increased activation status of T cells in vaccinated T1D patients may be associated—directly or indirectly—with improved glucose metabolism, contributing to better glycemic control.

Increased glucose consumption through increased glycolysis has been suggested as another mechanism leading to improved blood glucose control after BCG vaccination during overt T1D. Specifically, it is assumed that the BCG-induced reduction in blood glucose levels is supported by a metabolic shift towards increased aerobic glycolysis and reduced oxidative phosphorylation [11,27]. We found an increase in two high-energy intermediates of glycolysis, phosphoenolpyruvate (PEP) and 2/3-phosphoglycerate (2/3-PG). The increase of these two late-stage intermediates of glycolysis may indicate a preferential switch of glucose metabolism to glycolysis in vaccinated T1D patients. However, we could not detect differences in pyruvate/lactate and purine levels as described in T1D patients who received BCG vaccination during overt diabetes [11]. The measurements of lactate and pyruvate in non-fasting blood could play a role here, as these—in contrast to the other metabolites—correlate only to a lesser extent with fasting blood measurements [50]. A switch from oxidative phosphorylation to preferred aerobic glycolysis, also called the Wartburg effect, has been shown to play a role in several diseases. Initially described for tumor cells [51], it also could be confirmed in diseases like multiple sclerosis [52] and Alzheimer’s disease [53]. It has also been reported in response to mycobacterial infection [54] and an association with BCG vaccination has been postulated beyond T1D [55]. The exact molecular mechanism behind BCG-induced glycolysis is still under investigation. It may involve the upregulation of glycolysis-related genes and a shift in cellular metabolic programming to optimize carbohydrate metabolism [27,56]. For T cells in general, it has been shown that glycolysis strongly increases during activation [31]. This is accompanied by increased expression of multiple glucose transporters from the Glut family members and involvement of the mTOR- pathway [31], which also plays a role in BCG-mediated induction of glycolysis [55].

Taken together, the increased activation status of T cells after vaccination may go hand in hand with increased glucose consumption and glycolysis associated with better glycemic control in T1D patients. Future studies are needed to fully characterize the T cell phenotype and function following BCG vaccination and the mechanism behind its impact on glycemic control in T1D patients.

## Figures and Tables

**Figure 1 vaccines-12-00452-f001:**
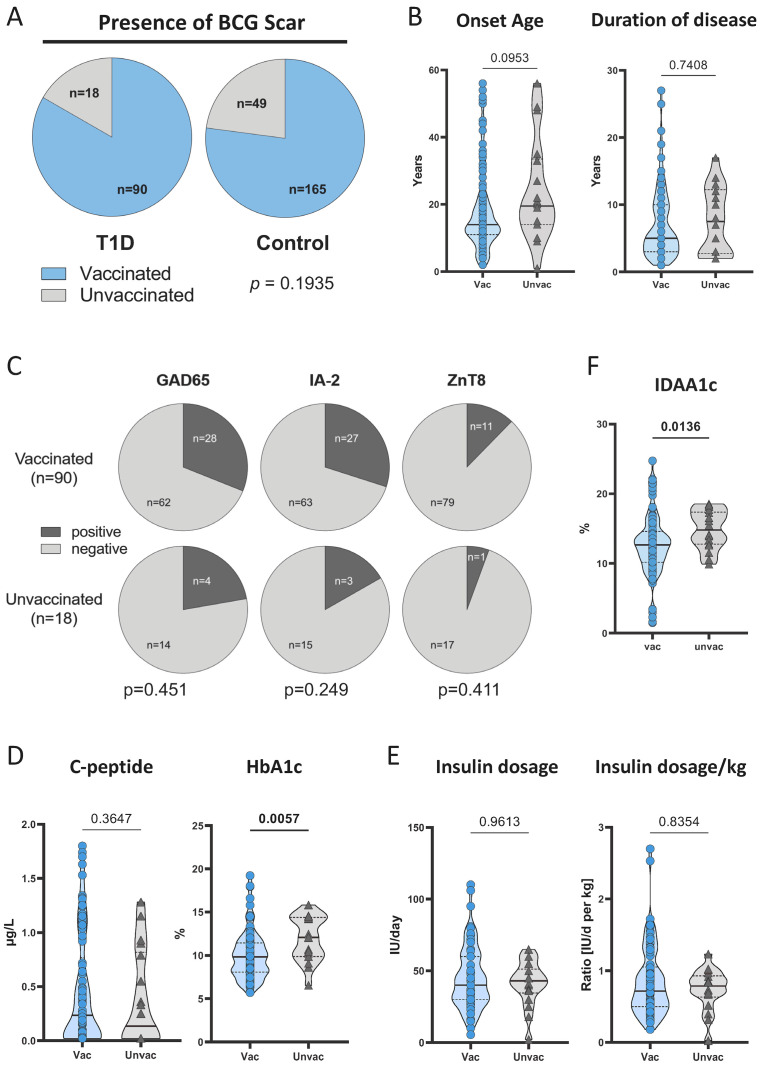
BCG vaccination rates and clinical characteristics of study groups. (**A**) The proportions of BCG-vaccinated individuals within the T1D population (*n* = 108) and healthy control population (*n* = 214) are shown in pie charts. The Fisher exact test was applied to analyze the significance of differences between the two study groups. The onset age and duration of disease (**B**) were compared between BCG-vaccinated (circles, *n* = 90) and unvaccinated (triangles, *n* = 18) T1D patients and are shown as violin plots. (**C**) The proportions of autoantibody positivity are shown as pie charts for the study groups of BCG-vaccinated (*n* = 90) and unvaccinated (*n* = 18) T1D populations. The Fisher exact test was applied to analyze the significance of differences between the two study groups. Comparisons of C-peptide and HbA1c (**D**), insulin dosage (**E**), and IDAA1c (**F**) between BCG-vaccinated (circles, *n *= 90) and unvaccinated (triangles, *n *= 18) T1D patients are shown as violin plots. Symbol plots including median values as straight lines and interquartile percentiles as dotted lines are given. Each symbol represents the mean of duplicates from an individual donor. Study group comparisons were performed, and *p*-values were calculated using the two-tailed Mann–Whitney U-test. Nominal *p*-values are given.

**Figure 3 vaccines-12-00452-f003:**
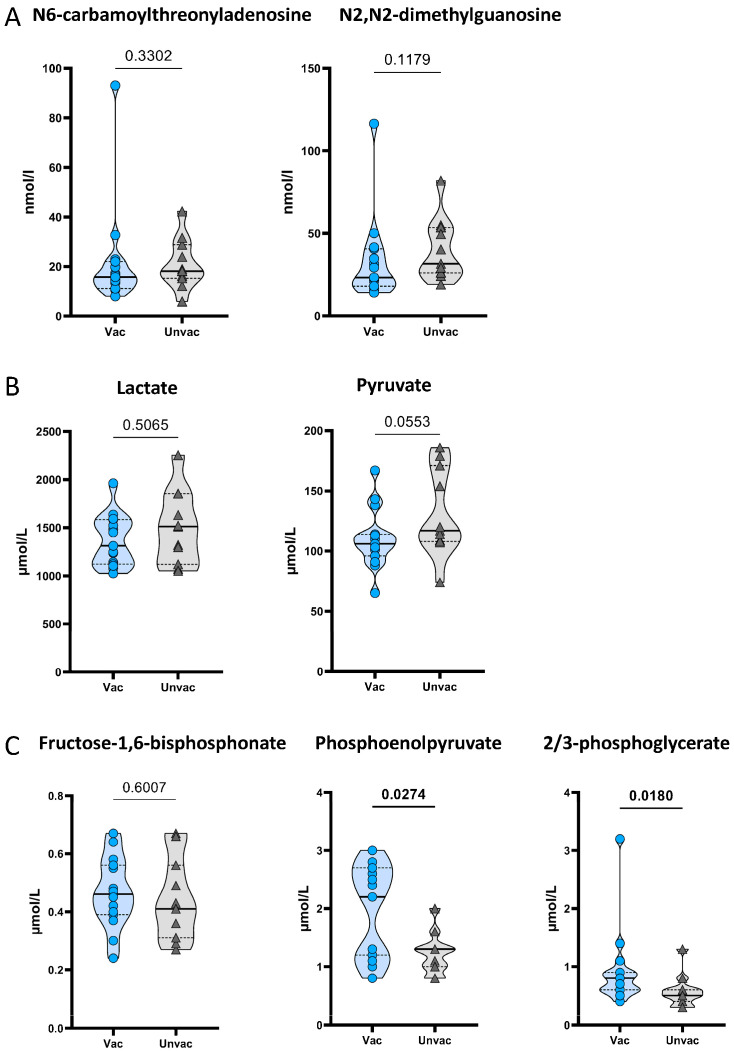
Metabolic comparisons of the relative levels of intermediates of glucose metabolism and purine synthesis between BCG-vaccinated and unvaccinated T1D patients. Mass spectrometric quantification of serum metabolites from BCG-vaccinated T1D patients (circles, *n* = 17) and unvaccinated T1D patients (triangles, *n* = 17) is shown for (**A**) intermediates of purine synthesis, (**B**) metabolic products of glycolysis, and (**C**) intermediates of glycolysis. Symbol plots including median values as straight lines and interquartile percentiles as dotted lines are given. Each symbol represents the mean of duplicates from an individual donor. Study group comparisons were performed, and *p*-values were calculated using the two-tailed Mann–Whitney U-test. Nominal *p*-values are given.

## Data Availability

Dataset available on request from the authors.

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
