# Peer review of "BCG Vaccination-Associated Lower HbA1c and Increased CD25 Expression on CD8+ T Cells in Patients with Type 1 Diabetes in Ghana"

_vaccines, 2024, doi:10.3390/vaccines12050452_

Round 1
Reviewer 1 Report
Comments and Suggestions for Authors
There are some more papers that probably should be cited that support the premise that BCG neonatal vaccines will prevent the onset of the disease and/or delay its onset.
Airaghi L, Tedeschi A. Negative association between occurrence of type 1 diabetes and tuberculosis incidence at population level. Acta Diabetol. 2006. https://doi.org/10.1007/s00592-006-0210-x PMID: 16865328
Kondrashova A, Reunanen A, Romanov A, Karvonen A, Viskari H, Vesikari T, et al. A six-fold gradient in the incidence of type 1 diabetes at the eastern border of Finland. Ann Med. 2005. https://doi.org/10. 1080/07853890410018952 PMID: 15902849
Doupis J, Markopoulou E, Efthymiou V, Festas G, Papandreopoulou V, Kallinikou C, et al. 55 th Annual Meeting of the European Diabetes Epidemiology Group of the EASD Scientific Committee P6 THE ROLE OF PEDIATRIC BCG VACCINE IN TYPE 1 DIABETES ONSET. 2021.
Corsenac P , Parent ME ́ , Mansaray H, Benedetti A, Richard H, Sta ̈ ger S, et al. Early life Bacillus Calm- ette-Guerin vaccination and incidence of type 1, type 2, and latent autoimmune diabetes in adulthood. Diabetes Metab. 2022; 48: 101337. https://doi.org/10.1016/j.diabet.2022.101337 PMID: 35245655
Dias, HF. Bacille CalmetteGuerin (BCG) and prevention of types 1 and 2 diabetes: Results of two observational studies. PlosOne 2023: https://doi.org/10.1371/journal.pone.0276423
Karaci M. The Protective Effect of the BCG Vaccine on the Development of Type 1 Diabetes in Humans. The Value of BCG and TNF in Autoimmunity. 2014. https://doi.org/10.1016/B978-0-12-799964-7. 00004–1
Reviewer 2 Report
Comments and Suggestions for Authors
Aniagyei et al. studied BCG vaccination in patients with type 1 diabetes in Ghana. They compared T1D-relevant clinical, immune, and metabolic parameters between matched BCG vaccinated (n=18) and unvaccinated (n=90) T1D patients. Lower HbA1c and IDAA1c levels in vaccinated T1D patients were accompanied by an increased CD25 expression on CD8+ T cells and a partial increase in the glycolytic intermediates phosphoenolpyruvate and 2/3-phosphoglycerate. The study provides some evidence that neonatal BCG vaccination may influence the clinical course of T1D. I have specific comments that should be addressed.
Major comments:
1. Since it has been shown that the number of BCG vaccinations could have different incidences of TD1, could you please specify if you know the number of BCG vaccinations received by the patients? Were multiple or single BCG vaccinations administered? Please comment on this point.
2. Do you know which BCG strain was used for the vaccination?
Minor comments:
1. In the abstract, the font changes in the 5th line starting at “Ghanaian long-term T1D…” Please ensure uniformity of the font.
Reviewer 3 Report
Comments and Suggestions for Authors
Vaccines-2953637
In the study of Aniagyei et al. the impact of BCG vaccination in Ghanaian T1D patients is tested. The author showed improved glycemic control and immune parameters. Patients with routine neonatal BCG vaccination had lower HbA1c levels compared to unvaccinated patients. CD8+ T cells from vaccinated patients showed higher CD25 expression and increased glycolysis metabolites.
Major Comments /Suggestions
1. Tests: The blood tests with non-fasting blood tests might be a source of high fluctuations. If this is not the case, please justify. This is important to assess the glycemic signature. Are the values of the components in Fig 3 affected (or not) by fasting blood tests?
2. Covid: T1D recruitment and tests were done during the era of COVID-19 (August 2021 to January 2023). It could be a critical yet uncontrolled variable. It will be important to include information on whether participants were sick with COVID and if so when on the time-line of T1D disease onset. The impact of the infection on metabolic regulation and glycemic control is well documented.
3. Time frame: Please add age distribution (as summarized in Suppl. Table S1). Most reported effects of BCG showed a clear decay from years of vaccination (5-15 years) for the mechanism of trained immunity. Data showing how many years passed from T1D onset to the blood measurements per person for the Van and Non vacc. is informative. There might be a bias that is dependent on the participants' age (especially ‘old’ age). The authors must address this time-line issue in view on the expected ‘decay’.
4. Statistics: While the statistic is well explained and justified, it is interesting to also analyze all metabolic components as a vector (and show a clustering/ distance in low dimensional space (PCA, clustering). This way, the statistics of different metabolic components show the person-to-person variability and will be easier to assess whether there is a partition of Vacc. vs nonVacc.
5. Biases: We appreciate that it is very hard to increase the number of the cohort and I appreciate the fact that it is hard to find T1D that were unvaccinated. However, selection bias is always a ‘tricky’ issue and hard to account for. Please specify how the people were recruited, what was the success rate for consenting?
There could be additional cofounders that are hard to control - for example, in the demographic data females may be pregnant or those that gave birth are probably under ‘non-standard’ immunological state that could be of relevance. If the information is available, it is an important addition.
6. New tests: I wonder if it will be possible to have data on the participants for another neonatal vaccination (like mumps and measles). Repeating the main analyses with adifferent vaccins can provide a strong validation for the fact that the finding s are all BCG-dependent.
Minor Comments:
1. Please add to Table S1: T1D vs healthy control (BMI)
2. Please add (in legend) how many data points included in each Fig. 2D panel (Reg NonReg vs All).
3. Several critical references that suggest BCG and long-term protection are missing. 1. Doi: 10.3390/vaccines8030378 that indicated the role of BCG coverage for about 15 years. 2. Doi: 10.3389/fimmu.2022.953228 that presenting the inconsistency in BCG as protective vaccination against COVID-19
Round 2
Reviewer 2 Report
Comments and Suggestions for Authors
The authors have satisfactorily responded my concerns